# VmambaSCI: Dynamic Deep Unfolding Network with Mamba for Compressive Spectral Imaging

## ABSTRACT

Snapshot spectral compressive imaging can capture spectral information across multiple wavelengths in one imaging. The method, coded aperture snapshot spectral imaging (CASSI), aims to recover 3D spectral cubes from 2D measurements. Most existing methods employ deep unfolding framework based on Transformer, which alternately address a data subproblem and a prior subproblem. However, these frameworks lack flexibility regarding the sensing matrix and inter-stage interactions. In addition, the quadratic computational complexity of global Transformer and the restricted receptive field of local Transformer impact reconstruction efficiency and accuracy. In this paper, we propose a dynamic deep unfolding network with mamba for compressive spectral imaging, called VmambaSCI. We integrate spatial-spectral information of the sensing matrix into the data module and employs spatial adaptive operations in the stage interaction of the prior module. Furthermore, we develop a dual-domain scanning mamba (DSMamba), featuring a novel spatial-channel scanning method for enhanced efficiency and accuracy. To our knowledge, this is the first Mamba-based model for compressive spectral imaging. Experimental results on the public databases, CAVE and KAIST, demonstrate the superiority of the proposed VmambaSCI over the state-of-the-art approaches.

## CCS CONCEPTS

• **Computing methodologies → Reconstruction**.

## KEYWORDS

Compressive Spectral Imaging, Deep Unfolding, Mamba

## 1 INTRODUCTION

Snapshot spectral compressive imaging offers significant advantages over traditional spectral imaging methods, including low cost, high speed, and minimal resource consumption. This technique finds widespread application in remote sensing, medical imaging, and various other fields. An exemplary method in this domain is Compressive Sensing Snapshot Spectral Imaging (CASSI) [39], which utilizes a coded aperture in conjunction with a dispersive prism to capture spectral information of a scene. Through the acquisition of a multiplexed 2D projection of the 3D data cube, CASSI

Permission to make digital or hard copies of all or part of this work for personal or classroom use is granted without fee provided that copies are not made or distributed for profit or commercial advantage and that copies bear this notice and the full citation on the first page. Copyrights for components of this work owned by others than the author(s) must be honored. Abstracting with credit is permitted. To copy otherwise, or republish, to post on servers or to redistribute to lists, requires prior specific permission and/or a fee. Request permissions from permissions@acm.org.
ACM MM, 2024, Melbourne, Australia
© 2024 Copyright held by the owner/author(s). Publication rights licensed to ACM.
ACM ISBN 978-x-xxxx-xxxx-x/YY/MM
https://doi.org/10.1145/nnnnnnn.nnnnnnn

facilitates the easy acquisition of spectral data from a single exposure. However, recovering a high-fidelity 3D hyperspectral image (HSI) from this 2D measurement poses a significant challenge.

Many approaches have emerged to tackle the ill-posed reverse problem of HSI reconstruction, ranging from traditional model-based approaches [3, 26, 41, 45] to learning-based approaches [32, 34]. The former leverages priors in different solution space with interpretability, such as non-local similarity [16], low-rank property [28], sparsity [49] and total variation [14]. However, the hand-crafted priors suffer from limited generalization, resulting in a mismatch with the problem. Besides, they often requires time-consuming numerical iterations. The latter introduces neural networks in reconstruction. End-to-end algorithms [44] restore the original HSI by brute-force mapping by learning spatial and spectral information, but ignore the working principles of CASSI system. Without the guidance of physical models, these methods also lack transparency. Plug-and-play algorithms [33] plug fixed pre-trained denoisers into traditional model-based methods. They fail to learn the specific mapping for HSI, therefore limiting the reconstruction performance. Among all the ways and means, deep unfolding networks [7, 22, 40] have exhibited superior performance among various approaches. This method tackles a *data subproblem* through convex optimization and a *prior subproblem* applying neural networks (denoiser), offering both the interpretability of model-based methods and the learning power of neural networks.

The *data subproblem* is closely tied to the degradation process. Methods for obtaining the degradation matrix fall into two categories. One directly employs the sensing matrix as the degradation matrix [29, 31, 40]. Nevertheless, due to phase aberration, distortion and alignment of the continuous spectrum, as well as photon and dark current noise, a gap exists between the sensing matrix and the degradation process. The other learns the degradation matrix using a neural network [22]. [12] adopts residual learning to derive the degradation matrix with reference to the sensing matrix. Besides, the compression of pixels at different positions in the 3D cube is agnostic in measurement, while existing algorithms ignore this pixel-specific degradation. In [25], a pixel-level adaptive recovery at different locations is introduced to solve this problem. They did not integrate precise spatial-spectral information into gradient descent but rather involved the sensing matrix in a relatively fixed manner during iteration. Additionally, the inter-stage interaction of the prior module is also inflexible. Most methods directly concatenate the features extracted from the previous stage with the input of the current stage, disregarding the spatial structure and distribution of the data. Consequently, we opt to explore a dynamic deep unfolding framework to concurrently address both problems.

By treating the regularization term as a denoising problem in an implicit manner, a denoiser, typically implemented as an end-to-end neural network, is trained for the *prior subproblem*. Due to

the strong capability in modeling the interactions of non-local regions, Transformer has been widely applied in most of the existing methods. However, it faces two main issues. On the one hand, the computational complexity of global Transformer is quadratic to spatial dimensions, which is unaffordable in some cases. One the other hand, the receptive fields of local transformer are confined to the window at a specific location, unsuitable for high precision reconstruction. Recently, structured state-space sequence models (S4) have emerged as a versatile architecture in sequence modeling. In terms of efficiency, they exhibit linear or near-linear scaling complexity with sequence length, making it particularly suitable for processing long sequences. In particular, the improved S4, known as Mamba, with selective mechanism and efficient hardware design, has been proven to surpass Transformer in tasks requiring long-term dependency modeling. More than that, some variants of Mamba have also been utilized in computer vision applications [23, 27, 51]. Motivated by this, we try to apply Mamba in hyperspectral snapshot compressive imaging. For better reconstruction, we start with the scanning mechanism to make full use of spatial information and spectral information.

Drawing from the above insights, we propose a dynamic deep unfolding network with visual state space model for compressive spectral imaging. Specifically, we introduce a flexible iteration strategy, enabling adaptable feature flow and fusion across each iteration. The proposed stage interaction module preserves and transmits important features according to the frequency and spatial characteristics of the current stage, which corrects the inherent information loss between stages and guides the following iteration. To address subproblems, we introduce a dynamic gradient descent module for data accuracy. Implementing a residual mapping, we align the sensing matrix with true degradation for result precision. Similar to stage interaction, we integrate current features into gradient descent for efficient optimization. Applying basic convolution and channel attention methods, we merge spatial and spectral information from the optimized sensing matrix for pixel-level degradation reconstruction. In addition, for the denoiser in the prior subproblem, we design a Mamba-based network architecture, called DSMamba. Different from scanning in spatial dimension in other frameworks, we develop a dual-domain scanning method, that is, scanning in both spatial and spectral dimensions at the same time, to integrate spectral information into spatial information. To alleviate Mamba's channel redundancy problem, we implement an efficient channel attention to select important channels to facilitate Mamba's learning of diversified channel representations. This supplementation compensates for the overall channel correlation lacking in DSMamba. By comparing the values of evaluation indicators and the quality of the reconstructed images with other existing methods, the proposed method achieves advanced performance.

The primary contributions presented in this paper can be summarized as follows:

- We devise a dynamic iteration strategy, which not only accords with the real degradation process, but also enhances adaptability to the features, thereby improving the reconstruction accuracy.
- We incorporate Mamba into the prior subproblem and propose a novel dual-domain scanning strategy to optimize

spatial-spectral information utilization. To the best of our knowledge, this is the first attempt to combine the physics-driven deep unfolding with Mamba in HSI reconstruction.

- We carry out extensive experiments on the simulation dataset. The proposed method, named VmambaSCI, outperforms state-of-the-art (SOTA) performance on HSI reconstruction.

## 2 RELATED WORKS

### 2.1 Deep Unfolding Networks for HSI Reconstruction

In general, when trying to reconstruct HSI, model-based techniques [1, 15, 24, 50] take a Bayesian view and treat it as an optimization problem that maximizes the posterior probability. The algorithms commonly used for optimization include HQS [20], ADMM [4], and PGD [2]. Normally, the techniques decouple the data fidelity and regularization terms in the objective functions resulting in the alternate solving of a data subproblem and a prior subproblem during iteration. The main idea of the deep unfolding [10, 38] is that the model-based iterative optimization algorithm can be implemented equivalently by deep neural networks. Originally applied to deep plug-and-play methods [30, 36, 46–48], this design utilizes a trained denoiser to implicitly represent a prior subproblem as a denoising problem. Inspired by this, deep unfolding methods are employed for specific tasks in an end-to-end training manner by jointly optimizing the trainable denoiser. For instance, GAP-net [31] unfolds the generalized alternating projection algorithm with a trained convolutional neural network. DGSMP [22] introduces an unfolding model estimation framework utilizing learned Gaussian-scaled mixed priors to improve performance. DAUHST [7] develop a novel half-shuffle Transformer to the unfolding framework. Addressing the gap between the sensing matrix and the real degradation, some approaches leverage neural networks to learn the latter. Due to the challenge of directly modeling the degradation process, RDLUF-MixS2 [12] applies residual learning to approach the degradation matrix. In PADUT [25], the pixel-specific degradation information is explored and the frequency information is introduced through a cross-stage fusion process. However, existing methods typically rely on simple concatenation for stage interaction. Therefore, it is necessary to propose a dynamic deep unfolding framework that can better accommodate degradation while enhancing and stabilizing feature interaction between stages, achieving the purpose of improving the network optimization process.

### 2.2 State Space Model

Recently, State Space Models (SSMs) have garnered attention. They not only establish long-distance dependencies but also exhibit linear complexity with respect to input size. The models rely on a classical continuous system mapping a one-dimensional input function or sequence $x(t) \in \mathbb{R}$, through intermediate implicit states $h(t) \in \mathbb{R}^{\mathbb{N}}$ to an output $y(t) \in \mathbb{R}$. The overall process can be expressed as a linear Ordinary Differential Equation (ODE):

$$
\begin{aligned}
h'(t) &= \mathbf{A}h(t) + \mathbf{B}x(t), \\
y(t) &= \mathbf{C}h(t) + \mathbf{D}x(t),
\end{aligned} \tag{1}
$$

where $N$ is the state size, $\mathbf{A} \in \mathbb{R}^{N \times N}$, $\mathbf{B} \in \mathbb{R}^{N \times 1}$, $\mathbf{C} \in \mathbb{R}^{1 \times N}$, $\mathbf{D} \in \mathbb{R}$.

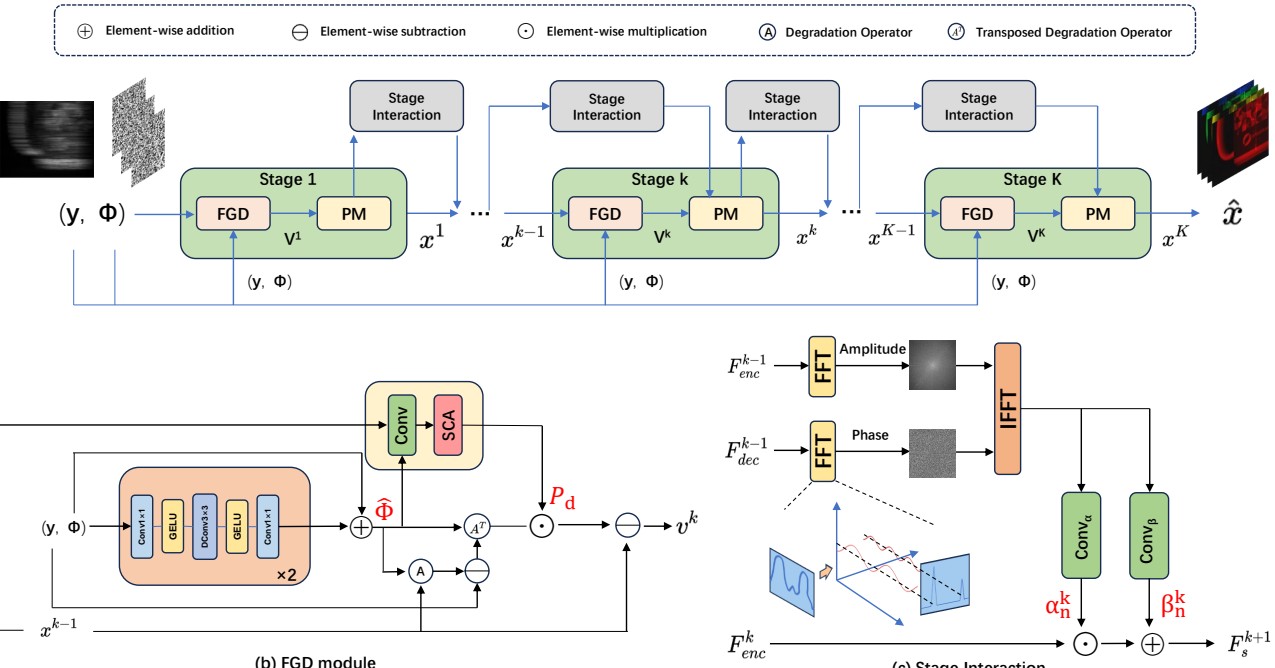

**Figure 1: (a) Overview of the proposed VmambaSCI with K stages. (b) The flexible gradient descent module (FGD). (c) The spatial adaptive frequency domain stage interaction.**

Discretizing the continuous system with the zero-order hold (ZOH) makes it more suitable for deep learning scenarios. Let $\Delta$ denote the timescale parameter to transform the continuous parameters $\mathbf{A}$, $\mathbf{B}$ to discrete parameters $\overline{\mathbf{A}}$, $\overline{\mathbf{B}}$:

$$
\begin{aligned}
\overline{\mathbf{A}} &= \exp(\Delta \mathbf{A}) \\
\overline{\mathbf{B}} &= (\Delta \mathbf{A})^{-1}(\exp(\mathbf{A}) - \mathbf{I}) \cdot \Delta \mathbf{B}.
\end{aligned}
\tag{2}
$$

After the discretization, the equations are written in linear recurrence form:

$$
\begin{aligned}
h_k &= \overline{\mathbf{A}} h_{k-1} + \overline{\mathbf{B}} x_k, \\
y_k &= \mathbf{C} h_k + \mathbf{D} x_k.
\end{aligned}
\tag{3}
$$

Besides, the above equations can be converted to CNN form:

$$
\begin{aligned}
\overline{\mathbf{K}} &\triangleq \left( \mathbf{C}\overline{\mathbf{B}}, \mathbf{C}\overline{\mathbf{A}}\overline{\mathbf{B}}, \cdots, \mathbf{C}\overline{\mathbf{A}}^{L-1}\overline{\mathbf{B}} \right) \\
\mathbf{y} &= \mathbf{x} \circledast \overline{\mathbf{K}},
\end{aligned}
\tag{4}
$$

where $L$ is the length of the input sequence, $\circledast$ represents convolution operation, and $\overline{\mathbf{K}} \in \mathbb{R}^L$ indicates a structured convolution kernel.

The recent advanced state space model, Mamba [17], improves $\overline{\mathbf{B}}$, $\mathbf{C}$ and $\Delta$ to be input-dependent, leading to a dynamic feature representation. By sharing the same recursive form of Eq. 3, Mamba can memorize ultra-long sequences, while the parallel scan algorithm [17] allows efficient training similar to the advantages in Eq. 4.

Due to Mamba's versatility, it has been extensively researched across various fields, including language understanding [18], general vision [27, 51], etc. Particularly, VMamba [27] brings Mamba to image classification tasks successfully. VM-UNet [35] utilizes a U-shaped network based on Mamba for medical image segmentation. VmambaIR [37] showcases the potential of state space models for Image restoration. Inspired by the above findings, we intend to apply Mamba to compressive spectral imaging. Although various Mambas typically scan in the spatial domain, considering the importance of spectral information in compressive spectral imaging, it is worthwhile to consider scanning the spectral dimension.

## 3 METHOD

### 3.1 Problem Formulation

Based on the compressive theory [13], the CASSI system can capture 2D measurements that contains information for all bands. The physical mask, denoted by $\mathbf{M} \in \mathbb{R}^{H \times W}$, can be viewed as a modulator which will act on the captured HSI signal $\mathbf{X} \in \mathbb{R}^{H \times W \times N_\lambda}$. Then the representation of the $n_\lambda^{th}$ wavelength of the modulated image:

$$
\mathbf{X}'_{n_\lambda} = \mathbf{M} \odot \mathbf{X}_{n_\lambda},
\tag{5}
$$

where $\odot$ represents the element-wise product.

After shifting along the horizontal direction according to the dispersion process, which is denoted as $d$, the signal can be expressed as:

$$
\mathbf{X}''(h, w, n_\lambda) = \mathbf{X}'(h, w + d_{n_\lambda}, n_\lambda),
\tag{6}
$$

where $\mathbf{X}'' \in \mathbb{R}^{H \times (W + d_{N_\lambda}) \times N_\lambda}$.

Ultimately, the imaging sensor captures the shifted image, and the final measurement can be formulated as:

$$\mathbf{Y} = \sum_{n_\lambda = 1}^{N_\lambda} \mathbf{X}''_{n_\lambda}, \tag{7}$$

where $\mathbf{Y} \in \mathbb{R}^{H \times (W + d_{N_\lambda})}$.

For the convenience, after taking the measurement noise into account, the matrix-vector form is formulated as:

$$\mathbf{y} = \Phi \mathbf{x} + \mathbf{n}, \tag{8}$$

where $\mathbf{x}$ is the original HSI, $\mathbf{y}$ stands for the 2D measurement, $\Phi$ denotes the sensing matrix that contains the whole degraded process, and $\mathbf{n}$ represents the additive noise. The purpose of HSI restoration is to recover the original high-quality image $\mathbf{x}$ from degraded measurement $\mathbf{y}$.

## 3.2 Dynamic Deep Unfolding Framework

The overall architecture of the proposed VmambaSCI method is depicted in Figure 1(a). It is a deep unfolding framework that composed of $K$ repeated stages. In each stage, there is a data module followed by a prior module. The former contains a Proximal Gradient Descent (PGD) algorithm which utilizes the physical degradation information, while the latter acts as a denoiser for optimization. In this section, we introduce the proposed dynamic stage interaction, which improves the quality of inter-stage fusion by transferring important information according to the current characteristics flexibly. The proposed data module and prior module are introduced in sections 3.3 and 3.4, respectively.

**Spatial Adaptive Frequency Domain Stage Interaction.** The inherent trade-off between spatial and spectral information leads to contextually different intermediate features at different stages of the encoder-decoder denoiser. Stage interaction can not only reduce the loss of information but also enrich the features of each stage. In [25], the frequency characteristics of HSIs are taken into account, which are ignored in the previous methods. In the encoder, the magnitude information is more prominent, which is concerned with the intensity of pixels. In the decoder, the phase information is more clear, which can help conveying positional information. However, this mode of feature transfer is not flexible enough. To address it, we introduce the normalization operation in a spatial-adaptive way, along with the frequency domain feature interaction. The process is shown in Figure 1(c). In this way, the denoiser can not only use more information of HSIs to guide the reconstruction but also reserve the refined memory of previous stages with the well-preserved spatial information, leading to an informative proximal mapping. The whole process can be expressed as:

$$\mathbf{H}_n^{k-1} = \mathcal{F}^{-1}\left(A\left(\mathcal{F}\left(\mathbf{F}_{enc}^{k-1}\right)\right), P\left(\mathcal{F}\left(\mathbf{F}_{dec}^{k-1}\right)\right)\right)$$
$$\alpha_n^k, \beta_n^k = \text{Conv}_\alpha\left(\mathbf{H}_n^{k-1}\right), \text{Conv}_\beta\left(\mathbf{H}_n^{k-1}\right) \tag{9}$$
$$\mathbf{F}_s^{k+1} = \mathbf{F}_{enc}^k \odot \alpha_n^k + \beta_n^k,$$

where $\mathcal{F}$ denotes the Fourier transform, $\mathcal{F}^{-1}$ represents the inverse Fourier transform. $A(\cdot)$ is the amplitude component, $P(\cdot)$ is the phase component, $\alpha_n^k$ and $\beta_n^k$ are tensors with spatial dimensions.

## 3.3 Flexible Gradient Descent

Mathematically, the optimization of HSI reconstruction can be modeled as:

$$\hat{\mathbf{x}} = \underset{\mathbf{x}}{\arg\min} \frac{1}{2} \|\mathbf{y} - \Phi \mathbf{x}\|_2^2 + \lambda J(\mathbf{x}), \tag{10}$$

where $J(\mathbf{x})$ denotes the regularizer term with parameter $\lambda$.

The PGD algorithm approximates Eq. (10) as an iterative convergence problem by the following iterative function:

$$\hat{\mathbf{x}}^k = \underset{\mathbf{x}}{\arg\min} \frac{1}{2\rho} \left\| \mathbf{x} - \left( \hat{\mathbf{x}}^{k-1} - \rho \Phi^T \left( \Phi \hat{\mathbf{x}}^{k-1} - \mathbf{y} \right) \right) \right\|_2^2 + \lambda J(\mathbf{x}), \tag{11}$$

where $\hat{\mathbf{x}}^k$ is the output of the $k$-th iteration, $\rho$ stands for the step size.

Actually, $\hat{\mathbf{x}}^{k-1} - \rho \Phi^T \left( \Phi \hat{\mathbf{x}}^{k-1} - \mathbf{y} \right)$ can be treated as a gradient descent operation and the rest can be solved by the proximal operator $prox_{\lambda, J}$. Thus, the reconstruction problem can be decomposed into two subproblems. One is the data module containing gradient descent, and the other is the prior module containing proximal mapping:

$$\mathbf{v}^k = \hat{\mathbf{x}}^{k-1} - \rho \Phi^T \left( \Phi \hat{\mathbf{x}}^{k-1} - \mathbf{y} \right),$$
$$\hat{\mathbf{x}}^k = \text{prox}_{\lambda, J}\left( \mathbf{v}^k \right). \tag{12}$$

For the gap between the sensing matrix and the real degradation, previous methods use neural networks to learn the degradation matrix,which is challenging. Later, estimating the residual between the sensing matrix and the degradation matrix from the 2D measurement and the sensing matrix has been proved to be more effective [12]. Meanwhile, a pixel-adaptive data module [25] was mentioned to address the problem of inconsistent and agnostic degradation at different locations in the HSI. Due to the lack of overall learnability of the gradient descent modules, they can not adjust themselves flexibly according to the current situation in each iteration, resulting in the reconstruction process deviating from the reality. Based on this, we propose a flexible gradient descent method that solves the problem of degradation more comprehensively, which is presented in Figure 1(b) . In addition to using residual learning to reduce the difference between the sensing matrix and the degradation matrix, we change the step size flexibly during the gradient descent. Instead of making it as a simple learnable parameter, we introduce spatial and spectral information carried by the degradation matrix to recover the input at the pixel level. In this way, the direction of the iteration remains correct.

Therefore, the gradient descent step in our proposed method can be expressed as:

$$\mathbf{v}^k = \hat{\mathbf{x}}^{k-1} - P_d \hat{\Phi}^{\top k} \left( \hat{\Phi}^k \hat{\mathbf{x}}^{k-1} - \mathbf{y} \right), \tag{13}$$

where $\hat{\Phi}^k$ denotes the calculated degradation matrix, $P_d$ represents the 3D parameters obtained by operating on input and degradation matrix, k is the number of stage.

## 3.4 Mixing Priors based on Mamba

As shown in Figure 2(a), the denoiser adopts a U-shaped structure consists of several basic unit blocks called MPMB. For the downsample layer, we use the convolution with a kernel size of 2 and a stride of 2. For the upsample layer, we use a pointwise convolution followed by a pixel shuffle operation. The block interaction

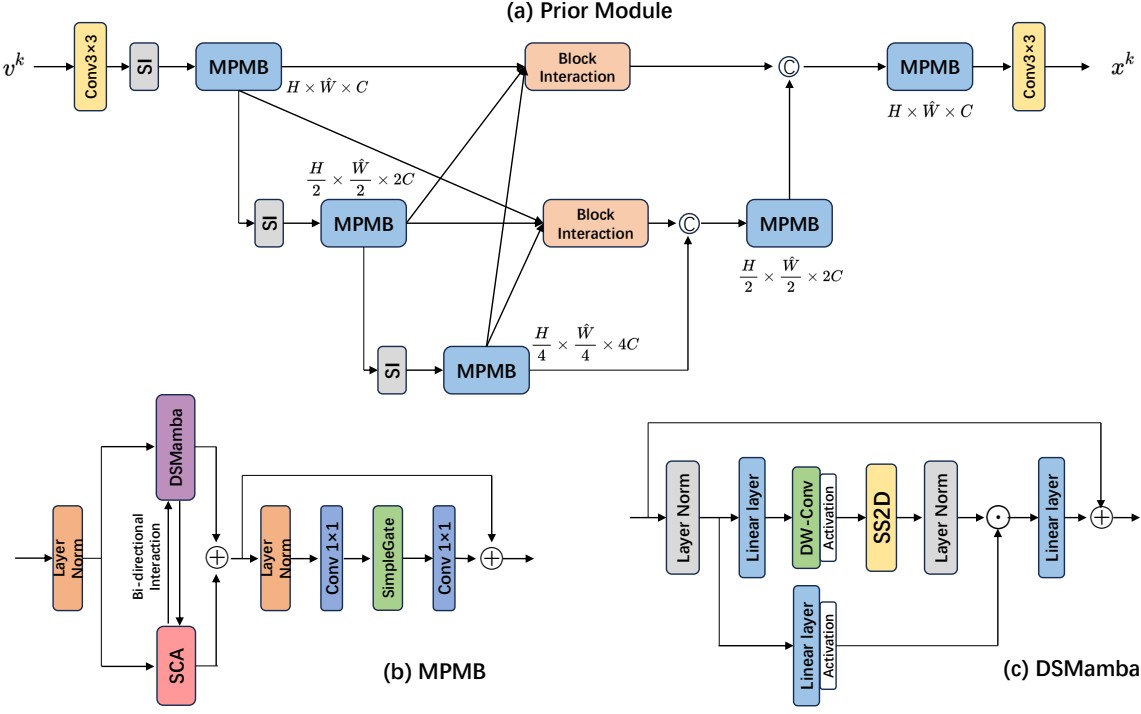

Figure 2: (a) PM is a three-layer U-shaped architecture with block interaction, SI represents stage interaction. (b) MPMB has two parallel branches, SCA and DSMamba, with a bi-directional interaction. (c) The core module of DSMamba is SS2D, in which we introduce spatial-spectral scanning.

is introduced to reduce the loss of information caused by sampling. Although SSMs introduce a large number of hidden states to memorize long-range dependencies, there exists notable channel redundancy [19]. What's more, the computational complexity of SSM is almost determined by the number of input channels. Considering the two points, unlike other methods of mapping features to higher dimensions, we set the channels of each layer in the network as 28, 56, 112.

**Mixing Priors based on Mamba Block.** The architecture of the MPMB is shown in Figure 2(b), which mainly consists of a spatial-spectral scanning mamba branch and a simplified channel attention branch in a parallel design. We add bi-directional interaction [12] between two branches to provide complementary clues in the channel and spatial dimensions. Instead of using gated-Dconv feed-forward network (GDFN) which often appears in other networks, we adopt the SimpleGate module [9] to reduce intra-block complexity while achieving better performance. The details of the two branches are described following.

**Spatial-Spectral Scanning Mamba Branch.** SS2D is the core operation of Mamba, which consists of three components: a scan expanding operation, a S6 block, and a scan merging operation. Derived from Mamba [17], the S6 block enables the model to distinguish and retain relevant information while filtering out the irrelevant by introducing a selective mechanism on top of S4 that adjusts the SSM's parameters based on the input. Unlike other

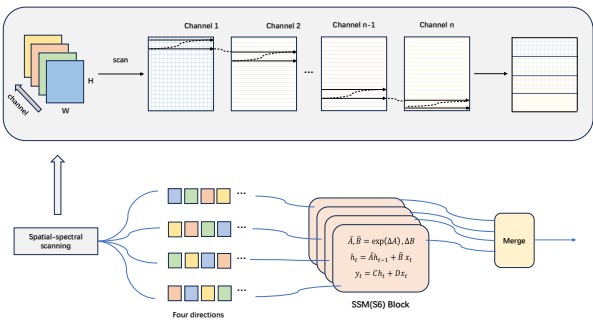

Figure 3: The flow of the SS2D operation. Above it there is an example of the space-spectral scanning.

methods which scans images in four directions of space (top-left to bottom-right, bottom-right to top-left, top-right to bottom-left, and bottom-left to top-right) into sequences, we add the scanning of the channel dimension as well. As shown in Figure 3, we take the first few lines of the first channel, the next few lines of the second channel, all the way to the last few lines of the last channel. To obtain comprehensive spatial information in different spectra, the number of times we repeat this operation is the number of channels. The first time it goes from the first channel to the last channel, the second time it goes from the second channel to the last channel

and back to the first channel, and then the last time it starts from the last channel and then goes from the first channel to the penultimate channel. The data obtained after the operations are integrated and scanned using the four directions mentioned above. In order to balance performance and efficiency, we only use this type of scanning on the first layer of the U-shaped architecture. By fusing the features of the two dimensions, we can capture the relationship between different spectra while obtaining spatial information.

**Simplified Channel Attention Branch.** While DSMamba introduces spectral information, it uses only a portion of each channel's features at each scan, which is incomplete. To learn the weight of each channel and better capture the correlation between different channels, we build another branch. Meanwhile, the channel attention can help focus on learning diverse channel representations, which improves the channel redundancy mentioned above. For simplicity, we use an architecture based on convolution to achieve this goal. In [9], it has been proved that the nonlinear activation functions can be replaced by multiplication or removed, which simplifies the model without performance degradation. Following this, we introduce simplified channel attention(SCA) and SimpleGate to extract channel information. SCA can be formulated as:

$$SCA(\mathbf{X}) = \mathbf{X} * W \operatorname{pool}(\mathbf{X}), \tag{14}$$

where pool indicates the global average pooling, $W$ represents a fully-connected layer and $*$ is a channelwise product operation. SimpleGate can be expressed as:

$$\operatorname{SimpleGate}(\mathbf{L}) = \mathbf{L}\left[:,:,:\frac{C}{2}\right] \odot \mathbf{L}\left[:,:,\frac{C}{2}:\right], \tag{15}$$

where $\mathbf{L}\left[:,:,:\frac{C}{2}\right]$ and $\mathbf{L}\left[:,:,\frac{C}{2}:\right]$ represent the first half and the second half of the input L across channel, respectively.

Similarly, we can replace the gating mechanism used in previous works with SimpleGate, which improves the information flow through the network in a simpler way.

## 3.5 Loss Function

Instead of L1 loss and L2 loss which may cause the images to be too smooth and lack sensory realism, we adopt Charbonnier Loss, a stable loss function, in the reconstruction. It is expressed as:

$$\ell\left(\mathbf{I}', \hat{\mathbf{I}}\right) = \sqrt{\left\|\mathbf{I}' - \hat{\mathbf{I}}\right\|^2 + \epsilon^2}, \tag{16}$$

where $\epsilon = 10^{-3}$ is a constant.

In addition to calculating the losses of the reconstructed images with respect to the original HSIs, we also include the reversible loss [8]. Based on the nature of the reversible optical path, the reconstructions are projected back to the measurement space, and then the gaps between the projected data and the actual 2D measurements are calculated. By considering both forward and inverse losses, we can make the final results closer to the real hyperspectral images and thus improve the reconstruction accuracy. Therefore, we define the overall loss function as follows:

$$\mathcal{L} = \ell\left(\mathbf{x}_{out}, \mathbf{x}_{truth}\right) + \xi \cdot \ell\left(\mathcal{G}\left(\mathbf{x}_{out}\right), \mathbf{y}\right), \tag{17}$$

where $\mathbf{x}_{out}$ is the output of the network, $\mathcal{G}$ denotes the process of mask coding and dispersion, $\mathbf{y}$ represents the measurement of the CASSI system. $\xi$ stands for a penalty coefficient which is set to 0.2 by default.

## 4 EXPERIMENTS

Experimental setup, implementation details and result analysis will be introduced in this section.

## 4.1 Experimental Settings

We conduct experiments on the simulation dataset. Following the previous approaches [6, 22, 32], a set of 28 wavelengths ranging from 450-650nm, which are derived through spectral interpolation manipulation, are selected for HSIs.

**Simulation HSI Data.** For the simulation experiment, two widely used HSI datasets, CAVE and KAIST, are adopted. The former comprises 32 HSIs with a spatial size of $512 \times 512$, the latter consists of 30 HSIs with a spatial resolution at $2704 \times 3376$. Following prior works, we employ the CAVE dataset as the training set, while 10 scenes from the KAIST dataset are utilized for testing. During the training process, we apply a real mask with a size of $256 \times 256$.

**Evaluation Metrics.** The reconstruction performance of the methods are evaluated through peak signal-to-noise ratio (PSNR) and structural similarity index (SSIM) [43].

**Implementation Details.** During training, to get labels for the simulation experiment, the 3D HSI datasets are randomly cropped to generate patches of size $256 \times 256 \times 28$. Real shifted masks of dimensions $256 \times 310 \times 28$ are utilized at the same time. As for the data augmentation techniques, according to the previous works, we leverage random flipping and rotation. Our model is implemented using the PyTorch framework. Adam optimizer is adopted with hyperparameters $\beta_1 = 0.9$ and $\beta_2 = 0.999$. The training process spans 300 epochs in total, and cosine annealing scheduler with linear warm-up is utilized. The learning rate and the batch size are set to $2 \times 10^{-4}$ and 1, respectively.

## 4.2 Quantitative Results

In our study, we conduct a comprehensive analysis of the proposed VmambaSCI method and SOTA techniques. The techniques including traditional the model-based methods: TwIST [3] and DeSCI [28]; the end-to-end networks: HDNet [21], MST [6], CST [5]; the deep unfolding methods: RDLUF-MixS2 [12], PADUT [25] and DERNN-LNLT [11]. The effectiveness of the methods is evaluated by PSNR and SSIM, and the results from 10 simulated scenes are represented in Table 1. It can be seen that the methods based on deep unfolding are better than the other two methods. Specifically, compared to PADUT-12stg [25], RDLUF-MixS2-9stg [12], DERNN-LNLT-9stg [11], which represents the recent SOTA methods, the VmambaSCI-9stg outperforms them with improvements of 1.19 dB, 0.51 dB and 0.15 dB on average, respectively.

## 4.3 Qualitative Results

By using 3 of 28 spectral channels of a scene obtained from the simulation, we provide a comparison of the proposed VmambaSCI method with six SOTA methods. As can be observed in Figure 4, our method excels in producing visually smoother textures with more vivid edge details, while preserving the spatial information of the homogeneous regions. Specifically, the dynamic framework we used enhances the accuracy of feature extraction and feature flow, and Mamba-based branch effectively models long-range dependency

**Table 1: The PSNR (upper entry in each cell) in dB and SSIM (lower entry in each cell) results on 10 simulated scenes. The best results are in bold.**

| Algorithms | Scene1 | Scene2 | Scene3 | Scene4 | Scene5 | Scene6 | Scene7 | Scene8 | Scene9 | Scene10 | Avg |
|---|---|---|---|---|---|---|---|---|---|---|---|
| TwIST | 25.16 | 23.02 | 21.40 | 30.19 | 21.41 | 20.95 | 22.20 | 21.82 | 22.42 | 22.67 | 23.12 |
|  | 0.700 | 0.604 | 0.711 | 0.851 | 0.635 | 0.644 | 0.643 | 0.650 | 0.690 | 0.569 | 0.669 |
| DeSCI | 27.13 | 23.04 | 26.62 | 34.96 | 23.94 | 22.38 | 24.45 | 22.03 | 24.56 | 23.59 | 25.27 |
|  | 0.748 | 0.620 | 0.818 | 0.897 | 0.706 | 0.683 | 0.743 | 0.673 | 0.732 | 0.587 | 0.721 |
| HDNet | 35.14 | 35.67 | 36.03 | 42.30 | 32.69 | 34.46 | 33.67 | 32.48 | 34.89 | 32.38 | 34.97 |
|  | 0.935 | 0.940 | 0.943 | 0.969 | 0.946 | 0.952 | 0.926 | 0.941 | 0.942 | 0.937 | 0.943 |
| MST-L | 35.40 | 35.87 | 36.51 | 42.27 | 32.77 | 34.80 | 33.66 | 32.67 | 35.39 | 32.50 | 35.18 |
|  | 0.941 | 0.944 | 0.953 | 0.973 | 0.947 | 0.955 | 0.925 | 0.948 | 0.949 | 0.941 | 0.948 |
| CST-L | 35.96 | 36.84 | 38.16 | 42.44 | 33.25 | 35.72 | 34.86 | 34.34 | 36.51 | 33.09 | 36.12 |
|  | 0.949 | 0.955 | 0.962 | 0.975 | 0.955 | 0.963 | 0.944 | 0.961 | 0.957 | 0.945 | 0.957 |
| RDLUF-MixS2-9stg | 37.94 | 40.95 | 43.25 | 47.83 | 37.11 | 37.47 | 38.58 | 35.50 | 41.83 | **35.23** | 39.57 |
|  | **0.966** | 0.977 | 0.979 | 0.990 | 0.976 | 0.975 | 0.969 | 0.970 | 0.978 | 0.962 | 0.974 |
| PADUT-12stg | 37.36 | 40.43 | 42.38 | 46.62 | 36.26 | 37.27 | 37.83 | 35.33 | 40.86 | 34.55 | 38.89 |
|  | 0.962 | 0.978 | 0.979 | 0.990 | 0.974 | 0.974 | 0.966 | 0.974 | 0.978 | **0.963** | 0.974 |
| DERNN-LNLT-9stg | 38.26 | 40.97 | 43.22 | **48.10** | **38.08** | 37.41 | **38.83** | 36.41 | 42.87 | 35.15 | 39.93 |
|  | 0.965 | 0.979 | 0.979 | **0.991** | **0.980** | 0.975 | **0.971** | 0.973 | 0.981 | 0.962 | **0.976** |
| Ours-9stg | **38.28** | **41.50** | **43.90** | 47.88 | 37.68 | **37.68** | **38.83** | **36.47** | **43.34** | 35.21 | **40.08** |
|  | **0.966** | **0.981** | **0.981** | 0.990 | 0.979 | **0.976** | 0.970 | **0.975** | **0.983** | 0.963 | **0.976** |

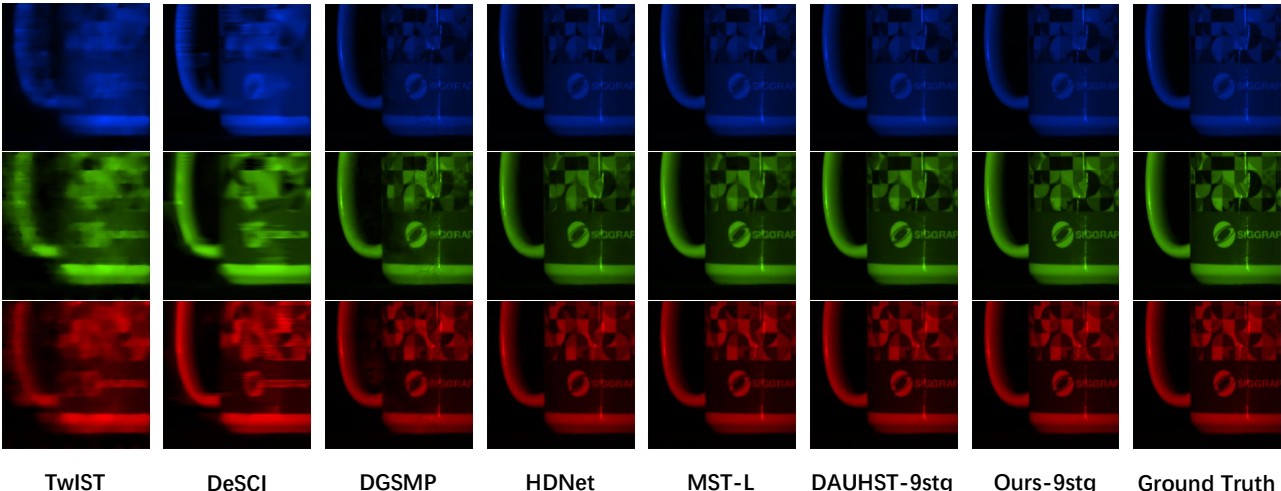

| TwIST | DeSCI | DGSMP | HDNet | MST-L | DAUHST-9stg | Ours-9stg | Ground Truth |

**Figure 4: Visual comparisons of reconstructed HSIs in scene 5 with 3 spectral channels.**

in two domains. Besides, Figure 6 illustrates the corresponding spectral density curves of different methods. Our method achieves the highest correlation coefficient, which indicating the spectral fidelity of our method.

## 4.4 Ablation Study

We conduct an ablation study to analyze the specific effects of different components of VmambaSCI on the overall performance, detailed in Table 2. First, we build a baseline model by adopting a

SCA branch and a convolutional self-attention branch [42] with a residual learning gradient descent module and a frequency domain stage interaction, achieving a result of 37.69 dB. Incorporating a pixel-adaptive gradient descent operation and spatial-adaptive normalization between stages increases the PSNR by 0.32 dB. Moreover, utilizing DSMamba yields an improvement of 0.69 dB, demonstrating significant enhancement. Combining these components results in a performance increase of 1.01 dB for VmambaSCI, affirming the

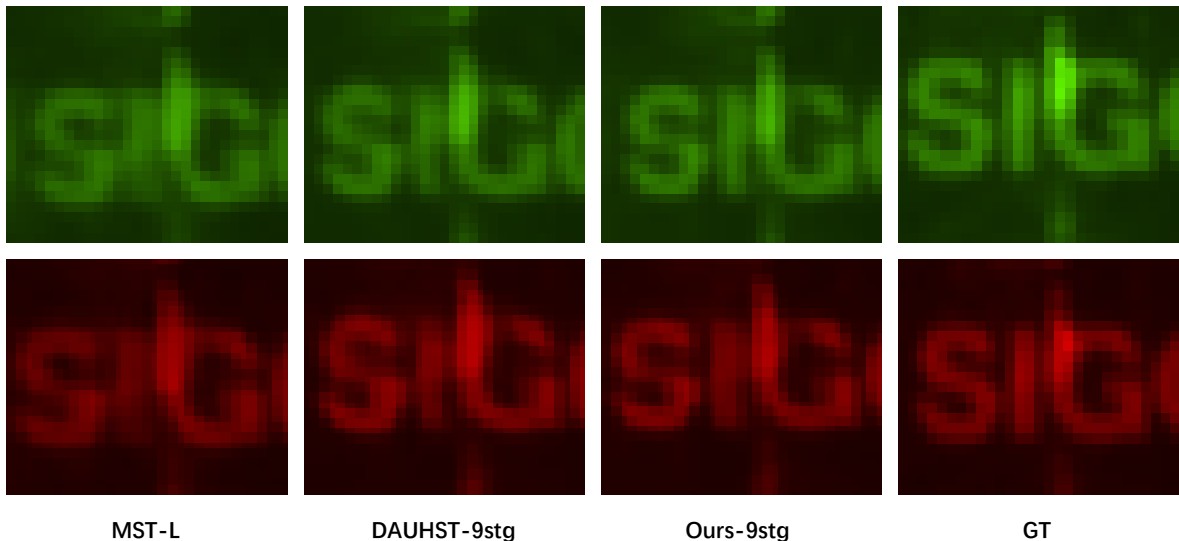

| MST-L | DAUHST-9stg | Ours-9stg | GT |

**Figure 5: Examples of the region which are chosen for the analysis of the reconstructed spectra. By looking at the images we can also find that the reconstructions obtained by our method are clearer and closer to the true values.**

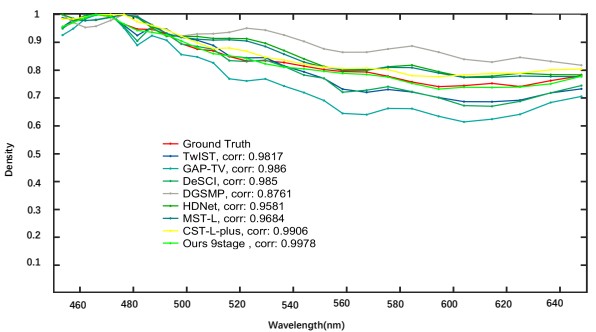

**Figure 6: Spectral Density Curves**

**Table 2: Break-down ablation study on individual components.**

| | | PSNR | SSIM |
|---|---|---|---|
| 1 | Baseline | 37.69 | 0.965 |
| 2 | 1 + Stage Interaction | 38.78 | 0.966 |
| 3 | 2 + Flexible Gradient Descent | 38.01 | 0.967 |
| 4 | 3+ DSMamba | 38.70 | 0.970 |

effectiveness of our method. Furthermore, we conduct an experiment to examine the impact of varying the number of stages in our model architecture, where parameters are shared except for the initial and final stages. Analysis of the findings presented in Table 3 reveals that as the number of stages increases (from 3 to 9), there is a corresponding enhancement in the network's performance. This underscores the efficacy of our iterative network

**Table 3: Ablation of number of stages.**

| Number of stages | PSNR | SSIM |
|---|---|---|
| 3 | 38.70 | 0.970 |
| 5 | 39.52 | 0.973 |
| 7 | 39.87 | 0.975 |
| 9 | 40.08 | 0.976 |

design, which enables refined information processing across multiple stages. Considering the trade-off between performance and computational complexity, the optimal configuration need to be determined for the specific task and constraints.

## 5 CONCLUSION

In this paper, we introduce a dynamic deep unfolding framework as an initial step, which not only narrows the gap between the sensing matrix and the degradation process, but also preserves more spatial information during feature fusion. To further enhance the spectral-spatial representation capabilities and model long-range dependencies without introducing excessive computational complexity, we propose the DSMamba, which scans in both spectral and spatial domains. By integrating the DSMamba into the framework, we create an end-to-end trainable neural network, referred to as VmambaSCI. Through comprehensive experiments, our proposed method achieves the best performance on simulated datasets. In future work, We will introduce our model to real datasets to examine the performance. Meanwhile, reducing memory costs and computation are worth noting. By building models that are easier to train and faster to infer, efficient techniques for HSI snapshot compression and reconstruction can be established.

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
