# OpenReview forum: "VmambaSCI: Dynamic Deep Unfolding Network with Mamba for Compressive Spectral Imaging"
_acmmm.org/ACMMM/2024/Conference — MM2024 Poster_

### Official Review · Reviewer_GXvN · 2024-05-03

**Rating:** 3
**Confidence:** 4

**Summary:**

This work presents VmambaSCI, a deep unfolding network tailored for snapshot compressive imaging (SCI) of hyperspectral images. In each proximal gradient descent (PGD) algorithm-unfolded stage, three key designs are implemented. Firstly, a flexible gradient descent (FGD) module is employed to dynamically learn element-wise step sizes for the gradient descent step. Secondly, the stage interaction (SI) module facilitates the transmission of high-capacity U-Net features between adjacent stages with different spatial scales. These features are fused in the frequency domain and utilized to modulate the U-Net encoder's features adaptively. Thirdly, a dual-domain scanning mamba (DSMamba) is devised to capture dependencies in both spatial and spectral dimensions while maintaining linear complexity. Experimental results demonstrate that VmambaSCI outperforms existing methods in SCI performance. Furthermore, the effectiveness of SI, FGD, and DSMamba is validated through ablation studies.

**Strengths:**

1. This is the first work, to my knowledge, utilizing vision mamba to address the problem of snapshot compressive imaging of hyperspectral images. The motivation behind reducing the quadratic complexity of transformers while maintaining a global receptive field is clear.
2. Experimental results demonstrate that the proposed method outperforms the previous best method by 0.15dB in PSNR.

**Limitations:**

1. The paper organization is poor, distracting readers from concentrating on the most important component—mamba. Unfortunately, a large portion of the text introduces details of micro-designs, which have already been investigated in previous works. For instance, ideas such as flexible gradient descent, stage interaction regarding multi-scale U-Net, FFT for feature modulating, simple gate, and simple channel attention have all been explored before [R1]-[R4]. Seeing an "A+B+C+D"-style work in 2024 is uninteresting to me. Moreover, considering the existing use of vision mamba for image restoration in MambaIR and VmambaIR, the technical novelty of this work seems limited without providing new insights into the specific scenarios of spectral SCI.

2. The presentation is poor. Firstly, it is quite odd to introduce the stage interaction in Section 3.2 before presenting the optimization model in Equation (10). A more natural approach would be to begin with Equation (10), followed by detailing gradient descent and proximal mapping in a top-down manner, respectively. The current order leaves me puzzled. Additionally, the logic from lines 463-499 is disjointed. It is confusing to start with discussing block information loss and then abruptly shift to describing SSM. This disrupts readability. Moreover, I believe the SS2D operation depicted in Figure 3 is the most significant contribution of this work, as it suggests improved scans of mamba in both spatial and spectral dimensions. However, the corresponding descriptions are overly brief and unclear, making it challenging to grasp its working principle.

3. The method is overly complex. In contrast, the ablation study is surprisingly simple. I'm uncertain about the source of the high performance. The loss function comprises a Charbonnier Loss term and a measurement consistency term, and the network's micro-designs differ from existing baselines. Consequently, it's unclear where the high performance originates. The ablation studies fail to convince me. Firstly, it's expected that introducing stage interaction would enhance reconstruction, given the numerous works demonstrating this. Secondly, the flexible gradient descent doesn't interest me much, as its performance improvement is already validated by works [R1] and [R3]. Thirdly, adding DSMamba should logically improve performance by increasing network depth, capacity, and receptive field, thereby enhancing representation ability and transformation capability. However, it's perplexing to see that flexible gradient descent results in a 0.77dB decrease in PSNR. I find the results unconvincing. Fourthly, increasing the stage number typically boosts network capacity and performance, which isn't particularly exciting. There's ample room for exploration. As a researcher in this field, I'm eager to understand how mamba works and how information is scanned and aggregated in spatial and spectral dimensions. Visualizing the effective receptive fields of different methods could provide valuable insights. Unfortunately, I haven't encountered any insightful results. There's significant potential to delve deeper into these studies. Simply presenting PSNR/SSIM results isn't compelling in 2024 for a paper submitted to a top-tier conference like ACM Multimedia.

4. The experiments lack control over variables such as parameter number, network depth, and computational cost. It's unclear whether the performance gain stems from the increase in these factors or from the proposed idea. More experiments maintaining a consistent scale of these factors should be conducted to clarify this.

5. Throughout this paper, it's unclear why using a mamba network is superior to utilizing a transformer network like MST and CST, which offer powerful global adaptive aggregation. Additionally, there's a lack of comparison regarding computational cost and inference time. As a result, I'm not convinced about adopting the proposed method for tackling SCI task. I recommend offering more in-depth insights focusing on the working principle of SS2D and its assistance in processing spatial and spectral features.

References:

[R1] G2-DUN: Gradient Guided Deep Unfolding Network for Image Compressive Sensing

[R2] Deep Generalized Unfolding Networks for Image Restoration

[R3] Pixel adaptive deep unfolding transformer for hyperspectral image reconstruction

[R4] Simple baselines for image restoration

**Suitability:**

3

---

### Official Review · Reviewer_nAJm · 2024-05-23

**Rating:** 3
**Confidence:** 4

**Summary:**

This paper embeds VMamba into deep unfolding framework for snapshot compressive imaging, integrates spatial-spectral information of the sensing matrix into the data module and employs spatial adaptive operations in the stage interaction of the prior module. Furthermore, a dual-domain scanning mamba  is proposed.

**Strengths:**

The PSNR of the proposed model is comparable to that of state-of-the-art methods, and embedding Mamba into deep unfolding networks is a reasonable idea.

**Limitations:**

The application of fast Mamba to the deep unfolding framework is reasonable, considering the potential speed limitations of the unfolding network with multiple recurrent stages. However, the paper lacks analysis or results regarding the computational complexity, which is an important aspect. The author should address this omission.

In the abstract, the author claims the development of a dual-domain scanning Mamba (DSMamba), featuring a novel spatial-channel scanning method for enhanced efficiency and accuracy. However, the spectral scanning pattern shown in Fig.3 appears to be the same as that in a referenced paper (https://arxiv.org/pdf/2404.18213), while the spatial scanning seems to be a special case of the current paper. The author is encouraged to clarify the differences to better highlight the contribution.

Another suggestion is that another contribution of this paper: the stage interaction, is also used in [1], to highlight the author’s contribution, it is suggested to compare the two stage interaction modules independently.

[1] Dong, Yubo, et al. "Residual degradation learning unfolding framework with mixing priors across spectral and spatial for compressive spectral imaging." Proceedings of the IEEE/CVF Conference on Computer Vision and Pattern Recognition. 2023.

**Suitability:**

2

---

### Official Review · Reviewer_xssC · 2024-05-25

**Rating:** 5
**Confidence:** 4

**Summary:**

This work proposes a new method, namely VmambaSCI, for spectral compressive imaging reconstruction. There are two main technical contributions:

(i) A dynamic deep unfolding framework that not only accords with the real degradation process but also enhances adaptability to the features.

(ii) A dual-domain scanning mamba (DSMamba) featuring a novel spatial-channel scanning method for enhanced efficiency and accuracy.

**Strengths:**

(i) The novelty is very good. It is a valuable exploration of Vision Mamba in SCI. It is exciting to see the technical breakthrough in this niche research topic.

(ii) The performance is solid. The proposed VmambaSCI outperforms the SOTA method. The visual results look good. The spectral curve also achieves the highest correlation.

(iii) The writing is clear and easy to follow.

**Limitations:**

(i) There are no qualitative results on the real dataset

(ii) More explanation and discussion between Transformer and Mamaba on the topic of SCI are required. Why Mamba is better and more suitable for SCI reconstruction than Transformer?

(iii) In table 1, Figure 4 and 5, the results of DAUHST, CST, MST, and BiSCI [1] should be added for a more comprehensive comparison.

[1] Binarized Spectral Compressive Imaging. In NeurIPS 2023.

**Suitability:**

3

---

### Official Review · Reviewer_DFin · 2024-05-26

**Rating:** 4
**Confidence:** 3

**Summary:**

In this paper, author propose a dynamic deep unfolding network
with mamba for compressive spectral imaging, whcih integrate spatial-spectral information of the sensing matrix into the data module and employs spatial adaptive operations in the stage interaction of the prior module.  Furthermore, author develop
a dual-domain scanning mamba (DSMamba), featuring a novel spatial-channel scanning method for enhanced efficiency and accuracy.

**Strengths:**

The author incorporate Mamba into the prior subproblem and propose a novel dual-domain scanning strategy to optimize, spatial-spectral information utilization. Extensive experiments show the better performance.

**Limitations:**

1.In the description of the paper, the motivation of the author's paper is not clear enough, and it only emphasizes the introduction of mamba results into in-depth development for the first time.
2.The author does not show the number of parameters in the model.

**Suitability:**

2

---

### Meta-Review · Area_Chair_qV5n · 2024-07-04

**Recommendation:** Accept (Poster)
**Confidence:** 5

**Metareview:**

The authors provided a response to the reviewers' concerns.

Reviewer nAJm points out that "the primary contribution of this paper is the application of Mamba to the snapshot task, which seems rather limited" and the other three reviewers while acknowledging this limitation are generally appreciative about the better performance and consider that the strengths outweigh the weaknesses in this work.

The Meta-Reviewer agrees with the reviewers that the paper has merits and that the authors' response was useful and its contents should be integrated together with the promised improvements in the camera ready paper and the codes should be released for reproducibility.